# FLM-Audio: Contiguous Monologues Improve Native Full-Duplex Chatbots via Dual Training

## Abstract

Full-duplex dialog models aim to listen and speak simultaneously, delivering rapid responses to dynamic user input. Among different solutions to full-duplexity, a *native* solution merges multiple channels in each time step, achieving the lowest latency. However, prevailing designs break down the textual monologue sentences for word-level alignment with audio streams, which degrades language modeling abilities. To help address this issue, we introduce "contiguous monologues", which are composed by continuous sentences and "waiting" intervals, mimicking human-like cognitive behavior in dialogs. We find a proper training paradigm to be critical for semantically aligning contiguous monologues with audio. To this end, we develop a "dual" training paradigm that alternates the position of the monologues, either leading or trailing the audio, across different training stages. A combination of our contiguous monologue and dual training strategy is applied in developing FLM-Audio, our 7B spoken dialog chatbot with native full-duplexity. As confirmed by experimental results, FLM-Audio achieves superior response qualities and chatting experiences while requiring significantly less training data.

## 1 Introduction

Human-like responsiveness is increasingly regarded as a key capability for applied AI systems. Human respond to rapidly-changing multimodal inputs with real-time speech, monologues, gestures, and actions. Therefore, achieving comparable responsiveness is recognized as a critical requirement for advanced AI, particularly for higher levels of embodied intelligence such as L3+ embodied AGI (Wang & Sun, 2025). In this paper, we focus on the audio and textual modalities, investigating human-like responsiveness with Spoken Dialog Models (SDMs). Such responsiveness is two-folds: it involves both human-like dialog behaviors (e.g., natural speech style, turn-taking, and graceful handling of interruptions) and human-like response latency (e.g., reacting promptly to dynamic environmental inputs). A common architectural principle underlying these behaviors is the implementation of full-duplex mechanisms (Lin et al., 2025; Zhang et al., 2024b; Wang et al., 2024a).

Two major strategies have emerged for full duplexity: *Time-Division Multiplexing* (TDM) and *Native Full-duplexity* (Figure 1). TDM, widely adopted in state-of-the-art audio-language models (Borsos et al., 2023; Zeng et al., 2024; Xie & Wu, 2024; Chu et al., 2023; Wang et al., 2024b), interleaves listening and speaking tokens within a single sequence. In each forward pass, a TDM model's context is a concatenated stream from all input and output channels (e.g., listening, monologue text, and speaking). As the Transformer attention mechanism (Vaswani et al., 2017) has a computational complexity of $O(n^2)$, TDM significantly hampers responsiveness, resulting in full-duplex delays of up to 2 seconds [1] (Zhang et al., 2024b), and limiting maximum generation length to roughly 45 seconds (Yuan Yao et al., 2025). These bottlenecks become increasingly restrictive as the foundation models continue to *scale up* (Hoffmann et al., 2022; Yao & Wang, 2023; Jaech et al., 2024).

On the other hand, the *Native Full-duplexity* approach (Figure 1, right), exemplified by Moshi (Défossez et al., 2024), tackles this scalability issue by merging all channels at each aligned time step, preventing the total context length from growing w.r.t. the number of channels, reducing the response

---

[1]This typically depends on the TDM chunk size, which can not be very small for semantic continuousness.

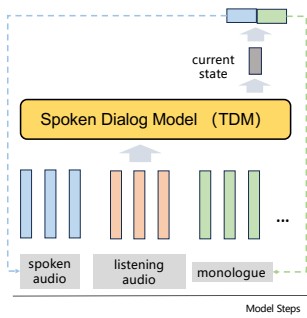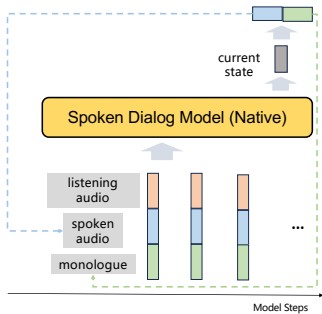

Figure 1: TDM vs. Native Full Duplexity for human-like responsiveness.

latency to as low as 80ms. However, aligning the textual monologue with the audio streams remains challenging due to the inherently different bitrates of each modality. In Moshi (Défossez et al., 2024), each monologue token is first generated in the text channel, and immediately pronounced in the speaking channel over the following time steps (typically 3∼4 steps). To accommodate this, monologue tokens are split by *<pad>* tokens to match the audio bit rate, waiting until the corresponding speech word is completed (Figure 2, left). This potentially breaks the language capability of pre-trained foundation models and degrades the ASR and instruct-following performances.

In this paper, we follow the *Native Full-duplexity* paradigm for its superior scaling potential, but instead introduce continuous monologue tokens, which we term "contiguous" monologues. Instead of temporally aligning every token to its pronunciation, we generate uninterrupted token sequences in the text channel (e.g., a full sentence or paragraph) while the speaking channel concurrently produces audio. Typically, the textual sentence finishes much earlier than the speaking channel due to different bit rates. During this gap, the model emits continuous *<wait>* tokens until the next monologue sentence is triggered. This approach preserves the language modeling strength of foundation models. Furthermore, during pre-training, transcripts and audio only need to be aligned at the sentence level, which both lowers pre-processing cost and mitigates error propagation from misaligned word timestamps. Figure 2 illustrates the contrast between alignment strategies.

Incorporating contiguous monologues in native full-duplex paradigm is a non-trivial problem: compared to word-level alignment, a model with contiguous monologues needs to learn to generate text and audio simultaneously, even when their semantic contents are asynchronous (e.g., the speech channel may still be pronouncing word A while the text channel has already advanced to words B or C). Our experiments show that the optimal stream arrangement, training objective, and configuration details differ substantially from those in related work (Défossez et al., 2024). To this end, we design a "dual" training scheme, where the contiguous monologue alternately leads or lags behind the audio channel across training stages, effectively covering both ASR- and TTS-like modes. We observe that such training strategy enables the model to handle the asynchronous semantics across long paragraphs, yielding coherent contiguous monologues and human-like natural speech at the same time.

The contributions of this paper include: (1) We propose a novel framework for native full-duplex audio chatbots, featuring a stream organization method based on contiguous monologues, as well as the corresponding complete training pipeline. (2) We release FLM-Audio, an open-source full-duplex audio-language model, along with the codes for the inference and interaction pipeline. Urls will be available upon publication. (3) Experimental results show that FLM-Audio outperforms native full-duplex baselines with much less post-training data, and surpasses state-of-the-art models in human-like responsiveness tests including automatic and human evaluation.

## 2  FLM-AUDIO: MODEL DESIGN

In this section, we introduce FLM-Audio, a native full-duplex model utilizing contiguous monologues through multi-stage dual training. FLM-Audio follows the *native* full-duplex approach (Figure 1, right), merging listening, speaking, and monologue channels at each autoregressive (AR) step of the backbone model. As discussed above, this approach avoids time-slice sharing by time-division multiplexing (TDM). We summarize previous work in Table 1, observing that most existing audio-

**Table 1:** Summary of related work. **Full-Duplex** stands for whether the model demonstrates capabilities to listen and speak simultaneously, with the minimal requirement of reacting promptly to interruptions in the listening channel. **E2E** denotes whether the model is end-to-end: an E2E model learns to directly generate audio tokens instead of relying on external ASR/TTS modules (though external token-to-wave audio decoders may still be used). Following Lin et al. (2025), we also summarize whether the full-duplex speech-to-speech pipeline is open-sourced (**S2S Release**).

| Method | Full-Duplex | Solution | E2E | S2S Release | Language |
|---|---|---|---|---|---|
| MiniCPM-Duplex (Zhang et al., 2024b) | ✓ | TDM | ✗ | ✗ | en |
| MiniCPM-Duo (Xu et al., 2024a) | ✓ | CDM | ✗ | ✗ | en |
| MinMo (Chen et al., 2025) | ✓ | TDM | ✓ | ✗ | multi |
| GLM-4-voice (Zeng et al., 2024) | ✗ | - | ✓ | - | en,zh |
| Kimi-Audio (Ding et al., 2025) | ✗ | - | ✓ | - | en,zh |
| Freeze-Omni (Wang et al., 2024b) | ✓ | TDM | ✗ | ✓ | en,zh |
| OmniFlatten (Zhang et al., 2024a) | ✓ | TDM | ✓ | ✗ | en,zh |
| Moshi (Défossez et al., 2024) | ✓ | Native | ✓ | ✓ | en |
| FLM-Audio (ours) | ✓ | Native | ✓ | ✓ | en,zh |

language models (as well as other omnimodal visual-language models such as MiniCPM-o (Yuan Yao et al., 2025) and Qwen2.5-Omni (Xu et al., 2025)) use TDM as a solution for full duplexity, with Moshi (Défossez et al., 2024) being a notable exception. While FLM-Audio adopts a similar backbone design to Moshi, we introduce key differences and improvements in stream organization, text–audio alignment, and the training pipeline.

## 2.1 BACKBONE STRUCTURE

Due to limitations in computational resources, we restrict the scale of our foundation model to ∼7B rather than using larger models such as Qwen-72B (Yang et al., 2025) and Tele-FLM-52B (Li et al., 2024). Since our goal is to support both English and Chinese, we also exclude English-only model families such as Llama (Meta, 2024). We opt to adopt a 7B-parameter autoregressive LLM as the backbone, initialized from the language model component of Qwen-2.5-VL (Bai et al., 2025)[2].

We follow the RQ-Transformer architecture (Yang et al., 2023; Zhu et al., 2024) employed by Moshi (Défossez et al., 2024) for streaming audio processing. This choice ensures better comparability, and we believe that in LLM-driven research, meaningful gains can stem directly from innovations in data organization, alignment strategies, and training paradigms, even when the core architecture is kept intact. Audio waveforms are discretized at 12.5 frames per second, with 8 audio tokens per frame. In each time step (1 frame), a *depth* transformer (Défossez et al., 2024; Yang et al., 2023; Zhu et al., 2024) takes the last-layer hidden states from the backbone model as input, and generates 8 audio tokens (1 semantic tokens followed by 7 acoustic tokens) in a locally autoregressive manner. Streaming Mimi encoder and decoder [3] serve as bridges between tokens and waveforms.

With $e$ denoting the embeddings of textual or audio tokens, the backbone model $F$ is defined as:

$$e_t = e_t^{\text{text}} \oplus \sum_{i=0}^{7} e_{t,i}^{\text{listen}} \oplus \sum_{i=0}^{7} e_{t,i}^{\text{speak}}, \tag{1}$$

$$h_t = F_\theta(\{e_0, \ldots, e_t\}). \tag{2}$$

We observe in experiments that the hidden state $h_t$ is sufficiently informative for textual, semantic, and acoustic generation. As a result, the depth Transformer can depend solely on the local $h_t$, without the need to re-aggregate $O(N^2)$ contextual information as required in the "talker-like" architectures employed in other related work like Qwen2.5-omni (Xu et al., 2025).

---

[2]We choose to use Qwen-2.5-VL to retain visual capability for program management purposes.
[3]https://huggingface.co/kyutai/mimi

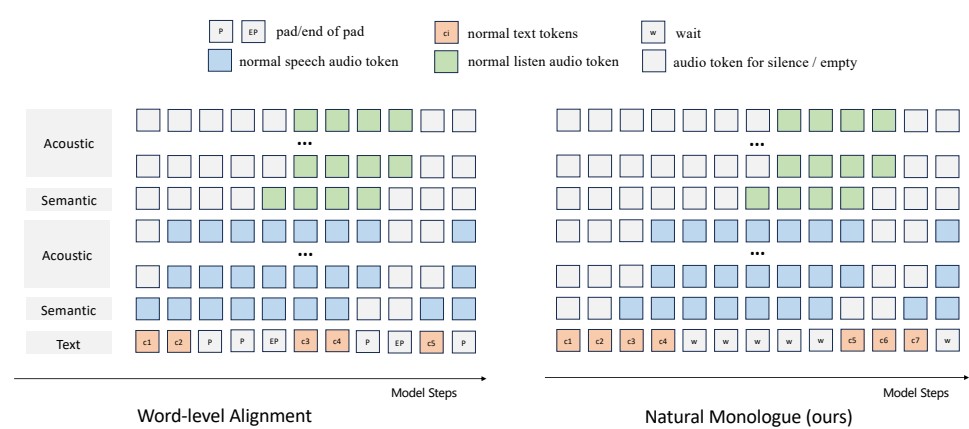

Figure 2: Stream organization for text and audio in FLM-Audio.

## 2.2 CONTIGUOUS MONOLOGUES

Even within a single utterance, textual and audio tokens are inherently asynchronous: one second of speech— represented by 12.5 frames of audio features—typically corresponds to only 3–4 monologue tokens. To address this mismatch, Moshi (Défossez et al., 2024) adopts a token-level alignment strategy, where textual tokens are split with special *<pad>* and *<end-of-pad>* tokens, ensuring each token to appear precisely at the time it is spoken (Figure 2, left). While effective, this approach has two major drawbacks: (1) it requires fine-grained, word-level timestamps for training annotations, which significantly increases data processing cost and introduces vulnerability to cascading alignment errors; and (2) it diverges from human-like dialog patterns. In natural conversations, humans think, listen, and speak concurrently, with internal monologues forming a coherent, forward-moving stream that generally *precedes* speech. From an empirical perspective, fragmenting sentences into isolated word-level tokens undermines the language modeling capacity of the backbone, as noticed in the original work (Défossez et al., 2024). Consistently, related work has also reported limited instruct-following performance for Moshi (Chen et al., 2025; Zhang et al., 2024a; Lin et al., 2025).

To overcome these limitations, FLM-Audio adopts a "contiguous monologues" strategy: instead of aligning text and audio at word-level, the monologues are represented as continuous token sequences, separated into sentences. Importantly, this setting mirrors human-like cognitive behaviors. The contiguous monologues can either lead or follow the spoken audio.

*Lead*: With monologue preceding the speaking channel by around 0∼2 tokens (TTS-style), FLM-Audio yields the same full-duplex latency as Moshi, as illustrated in Figure 2 (right). Once the monologue sentence finishes, the text channel is filled with <wait> tokens until the corresponding speech concludes or is interrupted by new input.

*Follow*: The monologue trails the listening channel, facilitating tasks such as sentence-level ASR.

Contiguous monologues requires only sentence-level transcripts for training, which drastically reduces annotation cost. Furthermore, it preserves the autoregressive language modeling capabilities of the pretrained backbone, supporting both natural dialog generation and responsive full-duplex speech.

## 3 FLM-AUDIO: DUAL TRAINING PARADIGM

Although both FLM-Audio and Moshi adopt a RQ-Transformer (Yang et al., 2023; Zhu et al., 2024) model architecture, *Moshi's training pipeline can not be trivially transferred to FLM-Audio.* This is due to fundamental differences in monologue alignment strategies, as discussed in Section 2.2. Because our framework incorporates both TTS-style and ASR-style data formats throughout post-training and fine-tuning, we term this approach the "Dual Training Paradigm". We summarize the distinctions across post-training and fine-tuning stages compared to Moshi in Appendix B.

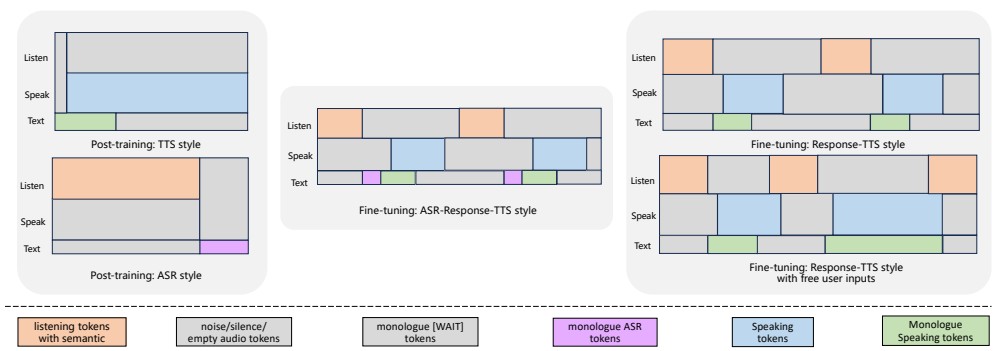

**Figure 3: Training data token organization in different stages**.

## 3.1 STAGE-1: POST-TRAINING

The objective of post-training is to equip the pretrained language model with both listening and speaking abilities. At this stage, large-scale audio-text data is used to train both autoregressive modeling of acoustic codes and semantic alignment between textual and audio modalities. For data processing, we compile a corpus of roughly 1 million hours of speech audio covering multiple Chinese and English sources including audio books, podcasts, TV shows, vlogs, etc. The audios are transcribed by FunASR (Gao et al., 2023) for Chinese and Whisper (Radford et al., 2023) for English, followed by text filtering to remove erroneous, harmful, or noisy samples.

The post-training has two sub-stages: in the first sub-stage (*Post-training-1*), the entire transcribed corpus is used as training data. In the second sub-stage (*Post-training-2*), we further incorporate a suite of open-sourced, human-annotated ASR datasets (Appendix A). To balance annotation quality, we down-sample the transcribed corpus from *Post-training-1* by half (as it relies on automatic transcripts) and up-sample the human-annotated datasets by a factor of 5 to emphasize their finer-grained accuracy. In both sub-stages, only sentence-level timestamps are extracted. Each aligned (audio clip, textual sentence) pair is tokenized and organized into two dual formats (Figure 3, left):

**TTS Style.** The listening channel is filled with empty tokens with all-0 embedding. The monologue is placed continuously on the text channel, while speech codes are filled into the speaking channel, beginning two tokens after the start of the text. Different aligned pairs are concatenated, separated by random silence, and padded to a uniform length of 8192.

**ASR Style.** The speaking channel is filled with empty tokens. Speech codes are placed on the listening channel, followed by the monologue text, effectively forming an ASR-style task.

In the post-training stage, we optimize a weighted cross-entropy loss over all non-empty tokens on the speaking channel, as well as all monologue and `<wait>` tokens on the text channel:

$$L = \alpha_1 * \text{CE}_{\text{speak, semantic}} + \alpha_2 * \text{CE}_{\text{speak, acoustic}} + \beta * \text{CE}_{\text{mono}} + \gamma * \text{CE}_{\text{wait}}, \tag{3}$$

in which $\alpha, \beta$, and $\gamma$ are tunable hyperparameters. The first audio token generated by the RQ-Transformer (channel index 0 for listen/speak) is the semantic token, while others are considered as acoustic tokens. We observed $\alpha_1 = 1, \alpha_2 = 0.5, \beta = 1, \gamma = 0.01$ to be effective.

This setup differs substantially from Moshi (Défossez et al., 2024), which reported optimal values of $\gamma = 0.5$ for their word-level `<pad>` tokens, $\alpha_1 = 100$, and $\alpha_2 = 1$. They also leveraged text-token masking and separate optimizers for the backbone and the depth model, whereas we observed such techniques to be unnecessary for training FLM-Audio.

Our stage-1 training corpus is approximately 1 million hours, considerably smaller than Moshi (8+ million hours) and other related work (Chen et al., 2025; Ding et al., 2025; Zeng et al., 2024). Nevertheless, FLM-Audio achieves comparable or superior performance in certain tasks (Section 4).

**Table 2: Training configuration for different stages.** The learning rate follows cosine schedules.

| Stage | Post-training-1 | Post-training-2 | Fine-tuning-1 | Fine-tuning-2 |
|---|---|---|---|---|
| Data Format Used | TTS+ASR | TTS+ASR | ASR-Response-TTS | Response-TTS |
| Num. Epochs | 3.3 | 1 | 2 | 6 |
| Batch Size | 256 | 256 | 256 | 256 |
| Learning Rate | 2e-4∼1e-5 | 1e-5∼8e-6 | 1e-5∼8e-6 | 8e-6∼7e-6 |

## 3.2 STAGE-2: SUPERVISED FINE-TUNING (SFT)

Supervised Fine-tuning (SFT) is applied to incorporate the capabilities for a general-purpose SDM. In FLM-Audio, we set up two sub-stages, including a semi-duplex *Fine-tuning-1*, followed by a final stage *Fine-tuning-2* (Tables 2, 6).

### 3.2.1 DATA COLLECTION

We construct SFT data in a fully synthesized pipeline:

**Transcript Collection.** We curate textual Chinese and English instruct-following data from open-source corpora, including Magpie (Xu et al., 2024b), Belle (Ji et al., 2023), Infinity-Instruct (Zhao et al., 2024; Li et al., 2025), WizardLM (Xu et al., 2023), and Ultrachat (Ding et al., 2023). User instructions are retained, while responses are refined using the DeepSeek-V3 (Liu et al., 2024) API. To ensure suitability for TTS, we enforce constraints on length, style, and the use of special symbols. Dialog lengths vary from 1 to 10 turns, mixing natural multi-turn conversations (e.g., Ultrachat) with synthesized single-turn instruct-following examples. In total, we sample 200K dialogs as transcripts for speech synthesis.

**Audio Generation.** We collect over 700 human voices, filter them based on DNSMOS (Reddy et al., 2022), and use the selected voices as references for a locally deployed Fishaudio TTS system (Liao et al., 2024). For each textual transcript, two distinct user voices are sampled, while the model's voice remains consistent across all dialogs. To improve robustness, we augment training audio with both environmental and speech noise. The processing details are provided in Appendix C.

### 3.2.2 SUB-STAGES

For SFT, we first introduce a semi-duplex transition stage, *Fine-tuning-1*, which integrates the TTS and ASR capabilities learned during post-training. The token streams are organized as follows:

**ASR-Response-TTS Style.** As illustrated in Figure 3 (middle), utterances are arranged in a semi-duplex manner. The model first processes the entire user instruction and immediately transcribes it into ASR tokens in the monologue channel. This span begins with a special `<asr>` token and terminates with an `<answer>` token. During the ASR phase, the speaking channel remains silent. Once the `<answer>` token is reached, the model generates a textual response, and, with a delay of 2 steps, produces the corresponding speech output (a TTS rendering of the response) on the speaking channel. Following Moshi, a 1-step offset is maintained between the semantic channel and the seven acoustic channels. This style of data effectively combines the TTS-style and ASR-style training signals from Stage-1, embedding both capabilities into each dialog instance and facilitating smooth transfer between post-training and SFT.

After this transitional stage, we proceed to the final stage, *Fine-tuning-2*, which uses the same dialog transcripts but reorganizes the textual and/or audio tokens:

**Response-TTS Style.** As shown in Figure 3 (top right), we remove the ASR text from the semi-duplex ASR-Response-TTS format, retaining only the response monologue. In this setting, the model is required to infer the user's intent directly from audio input and generate the appropriate textual and spoken responses. After this stage, FLM-Audio achieves a response latency equivalent to Moshi, while maintaining strong language modeling performance.

**Response-TTS Style with Free User Inputs.** As shown in Figure 3 (bottom right), this style enables full duplexity. Here, user utterances may occur at arbitrary time, potentially interrupting the model's response, forcing the model to learn realistic turn-taking. Specifically, when interrupted by meaningful user speech, the model must cut off both its monologue and speaking channels, falling silent within a short delay. Once the interruption ends, it resumes dialog generation, potentially addressing a new topic. To simulate this behavior, interruptions are introduced with probability 0.7, and the reaction delay is tuned to 0.5 seconds to avoid oversensitivity to short back-channels.

### 3.3 TRAINING CONFIGURATION

We summarize the training hyperparameter configuration in Table 2.

## 4 EXPERIMENTS

As discussed above, FLM-Audio features native full-duplex design with contiguous monologues, and a 4-stage training paradigm with dual formats for data organization. Thus, we focus on answering the following three research questions with experimental observations: **RQ1:** In native full-duplex systems, do contiguous monologues improve semantic understanding as hypothesized? **RQ2:** How effective is the dual data-format strategy across training stages, and how crucial is it to final performance? **RQ3:** How does FLM-Audio compare against state-of-the-art full-duplex chatbots in terms of responsiveness, speech quality, and dialog capability? To address these questions, we benchmark FLM-Audio against representative clusters of existing models and systems across three dimensions: audio understanding, audio generation, and duplex dialog performance. In addition, we conduct ablation studies under different training configurations to isolate the effects of contiguous monologues and dual-format supervision.

### 4.1 AUDIO UNDERSTANDING

We evaluate audio understanding through automatic speech recognition (ASR) and spoken question answering tasks. For ASR, we adopt word error rate (WER) as the primary metric, testing on both Chinese and English benchmarks, including Fleurs-zh (Conneau et al., 2022) and LibriSpeech-clean (Panayotov et al., 2015). While instruction-following with spoken input is addressed separately in Section 4.3, we also include LlamaQuestions (Nachmani et al., 2023) as a speech-based QA benchmark, reporting accuracy.

For comparison, we include Whisper-large-v3 (Radford et al., 2023), Qwen2-Audio (Chu et al., 2023), MinMo (Chen et al., 2025), and GLM-4-Voice (Zeng et al., 2024), all of which are specialized audio–language models, as well as GPT-4o (Hurst et al., 2024), a proprietary large-scale system.

Table 3 presents the results. After both post-training and supervised fine-tuning (SFT), FLM-Audio shows strong performance on Chinese ASR, surpassing specialized systems such as Qwen2-Audio on the Fleurs benchmark. On LlamaQuestions, FLM-Audio achieves accuracy comparable to other bilingual Chinese–English models, demonstrating that its textual knowledge remains well preserved throughout training.

We emphasize the comparison to Moshi (Défossez et al., 2024), the only other native full-duplex audio–language model. Despite being trained with less than 15% of Moshi's audio data and without fine-grained timestamps, FLM-Audio achieves superior performance: on LibriSpeech-clean, FLM-Audio yields significantly lower WER. Furthermore, whereas Moshi is specialized for English, more than half of FLM-Audio 's training data is Chinese, enabling broader multilingual coverage.

Finally, we note a pronounced improvement in Chinese ASR performance after the *Post-Training-2* stage. This aligns with our training setup, where *Post-Training-2* replaces coarse ASR annotations with high-quality, human-annotated Chinese transcripts. English ASR, by contrast, already performs competitively after *Post-Training-1* even without additional fine annotations, suggesting that our contiguous monologue design provides a key advantage for capturing audio semantics.

**Table 3: Audio understanding results.** We include ASR and audio question answering benchmarks. Different results for a same model come from different evaluation sources, potentially indicating different inference configurations.

| Model | Fleurs zh (WER ↓) | LibriSpeech clean (WER ↓) | LlamaQuestions (Acc. ↑) |
|---|---|---|---|
| GPT-4o Hurst et al. (2024) | 5.4 | - | 71.7 |
| Whisper-large-v3 Radford et al. (2023) | 7.7 | 1.8 | - |
| Qwen2-Audio Chu et al. (2023) | 7.5 | 1.6 | - |
| MinMo Chen et al. (2025) | 3.0 | 1.7 | 64.1 |
| Freeze-Omni Wang et al. (2024b) | - | 3.82 | 72 |
| OmniFlatten Zhang et al. (2024a) | - | 7.91 | - |
| GLM-4-Voice Zeng et al. (2024) | - | 2.8 | 50.0 (64.7) |
| Kimi-Audio Ding et al. (2025) | 2.69 | 1.28 | - |
| Qwen-2.5-Omni Xu et al. (2025) | 2.92 | 2.37 | - |
| Moshi | - | 5.7 | 43.7 (62.3) |
| FLM-Audio (Post-1) | 7.2 | 5.3 | - |
| FLM-Audio (Post-2) | 5.5 | 4.6 | - |
| FLM-Audio (SFT-1) | 5.4 | 3.2 | 56.3 |

## 4.2 AUDIO GENERATION

We assess audio generation performance using the Seed-TTS-en and Seed-TTS-zh benchmarks (Anastassiou et al., 2024a), following the standard evaluation protocols. Results are presented in Table 4.

While FLM-Audio is not explicitly optimized for high-fidelity voice cloning-and therefore does not surpass state-of-the-art TTS systems in similarity (SIM) scores-it achieves word error rate (WER) performance comparable to advanced, specialized TTS models such as Seed-TTS (Anastassiou et al., 2024b) and CosyVoice (Du et al., 2024). Moreover, its WER scores are also on par with those of general audio–language models, including GLM-4-Voice and MinMo.

**Table 4: Audio generation results.** We include WER and speaker similarity as metrics. Similarity scores (*) are computed using a model that has been lightly fine-tuned, following a straightforward data format that incorporates reference audio.

| Model | Seed-tts-en | | Seed-tts-zh | |
|---|---|---|---|---|
| | WER ↓ | SIM ↑ | WER ↓ | SIM ↑ |
| Seed-tts | 2.25 | 0.762 | 1.12 | 0.796 |
| Cosyvoice | 4.29 | 0.609 | 3.63 | 0.723 |
| Cosyvoice2 | 2.57 | 0.652 | 1.45 | 0.748 |
| GLM-4-Voice | 2.91 | - | 2.10 | - |
| Minmo | 2.90 | - | 2.48 | - |
| FLM-Audio (SFT-2) | 2.95 | 0.543* | 2.10 | 0.601* |

## 4.3 FULL-DUPLEX CHATTING

For LLM-based assistants, full-duplex chatting differs substantially from traditional text-based multi-modal instruction-following, particularly with respect to human preference. In instruction-following tasks, users often value detailed, elaborate responses, such as those required for programming or complex reasoning (Jaech et al., 2024; DeepSeek-AI et al., 2025). In natural spoken conversations, however, users typically prefer concise, summarized, or even intentionally evasive replies. To capture these differences, we conduct a comprehensive evaluation combining both automatic metrics and human judgment.

**Automatic evaluation.**    We construct a speech instruction-following test set using publicly available Chinese prompts formatted in the style of AlpacaEval (Li et al., 2023). Prompts are converted into audio using our TTS pipeline. DeepSeek-V3 (Liu et al., 2024) is employed as a reference model to assign quality scores (0–10 scale) by comparing candidate textual responses to ground-truth answers.

**Human evaluation.**    We run a double-blind comparison between FLM-Audio and Qwen2.5-Omni (Xu et al., 2025), a state-of-the-art streaming chatbot. Five human annotators rate multi-turn audio responses across four dimensions: (1) Helpfulness, standing for the informativeness and relevance of content; (2) Naturalness, for conversational tone and linguistic fluency; (3) Responsiveness, representing reaction speed to interruptions and dynamic user input; and (4) Robustness, which means stability under noisy real-world conditions. This benchmark shares the same spirit as (Lin et al., 2025), but is constructed in Chinese.

**Results.**    Table 5 summarizes the results. Compared with Qwen2.5-Omni, FLM-Audio delivers responses of comparable quality in terms of helpfulness, as confirmed by both automatic scoring and human ratings. More importantly, in dimensions that matter most for real-time interaction: naturalness, responsiveness, and robustness, FLM-Audio demonstrates a clear advantage. We attribute this to the model's native full-duplex design and the effectiveness of the dual training paradigm.

**Ablation study.**    We further compare against a baseline trained without the semi-duplex *Fine-tuning-1* stage (i.e., omitting ASR-style supervision). This variant shows a marked drop in instruction-following ability, underscoring the importance of retaining the dual data format in SFT. In particular, the ASR-style organization significantly strengthens audio understanding, validating the design of our training pipeline.

**Table 5: Full-duplex Chatting results.** Automatic and human evaluation results are included.

| Model | Instruct LLM-score↑ | Human Evaluation | | | |
|---|---|---|---|---|---|
| | | Helpfulness↑ | Naturalness↑ | Responsiveness↑ | Robustness↑ |
| Qwen-2.5-omni | 6.36 | **7.4** | 7.9 | 8.1 | 7.7 |
| FLM-Audio w/o SFT-1 | 4.59 | - | - | - | - |
| FLM-Audio SFT full | **6.58** | 7.2 | **8.2** | **8.8** | **8.0** |

## 4.4 ANSWERS TO RESEARCH QUESTIONS

We now revisit the research questions posed at the beginning of Section 4:

*RQ1: In native full-duplex systems, do contiguous monologues improve semantic understanding as hypothesized?* Yes. As shown in Section 4.1, FLM-Audio matches Moshi's performance after only the *Post-training-1* stage, despite being trained with less than 15% of Moshi's audio data. Since both models share the same RQ-Transformer backbone and rely on coarse third-party ASR transcripts at this stage, the performance advantage is best explained by our contiguous monologue design. Additional evidence comes from final instruct-following results: FLM-Audio reaches performance levels comparable to state-of-the-art systems such as Qwen-2.5-Omni, whereas Moshi has been reported to lag behind in this area (Zhang et al., 2024a).

*RQ2: How effective is the dual data-format strategy across training stages, and how crucial is it to final performance?* The TTS-style format is essential for any responsive full-duplex system, including both FLM-Audio and Moshi. Thus, the key question is whether the additional ASR-style format provides a measurable benefit. Results in Table 3 and Table 5 confirm that it does: models trained without ASR-style supervision show clear disadvantages in audio understanding and instruction-following, underscoring the importance of dual-format training.

*RQ3: How does FLM-Audio compare against state-of-the-art full-duplex chatbots in terms of responsiveness, speech quality, and dialog capability?* As demonstrated in Section 4.3, FLM-Audio achieves comparable overall response quality to Qwen-2.5-Omni, while delivering superior naturalness, responsiveness, and robustness in interactive settings. These results affirm that native full-duplex design, coupled with our dual training paradigm, enhances both the quality and the real-time usability of spoken dialog systems.

## 5 CONCLUSION

In this paper, we introduced contiguous monologues for native full-duplex audio-language models, together with a dual training pipeline that integrates ASR- and TTS-like capabilities. Building on this design, we developed FLM-Audio, a bilingual chatbot. Compared with the most related baseline Moshi, FLM-Audio achieves equivalent response latency while delivering substantially stronger language modeling performance with less data. It also outperforms TDM-based systems in dialog experiences. Constrained by data volume and computational resources, we have not yet scaled FLM-Audio to larger parameter counts—a direction where native duplex models could exhibit even greater advantages over TDM-based approaches. We hope this research will inspire further exploration into scaling native full-duplex architectures, both to push the performance upper bound of task-solving and to provide a more comprehensive comparison against TDM-based solutions.

## ETHICS STATEMENT

The data used to train FLM-Audio is obtained exclusively from publicly available sources or through commercial licenses. No unauthorized or private data has been included. As FLM-Audio is developed upon a foundation language model and refined through post-training, harmful contents could potentially be elicited from the released model despite the efforts made for safety. The generated contents by FLM-Audio do not represent the opinions of the authors or entities involved.

## REPRODUCIBILITY STATEMENT

We provide comprehensive details of the training configurations and data preprocessing pipelines in Section 3 and the Appendix. The model checkpoint and interactive interface will be publicly released. We hope these efforts facilitate the community in implementing our methods and achieving results consistent with our conclusions.

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

## A    Curated open-sourced ASR datasets

The datasets include ST_CMDS[4], Aishell3 (Shi et al., 2020), Magicdata[5], primewords_md_2018_set1 [6], and Thchs30 (Wang & Zhang, 2015).

## B    Training stage comparison with Moshi

We summarize different training stages compared to Moshi in Table 6. Both models undergo four stages in total, including two post-training and two fine-tuning stages. However, to better exploit the language modeling benefits of contiguous monologues, FLM-Audio features special designs to enhance sentence-level alignment with both listen and speak channels during the early stages.

**Table 6: Training Paradigm: FLM-Audio and Moshi.**

| Model | LLM | Post-training-1 | Post-training-2 | Fine-tuning-1 | Fine-tuning-2 |
|---|---|---|---|---|---|
| Moshi (Défossez et al., 2024) | Helium | 1-channel | 2-channel semi-duplex | full-duplex dialog | full-duplex instruct |
| FLM-Audio | Qwen-2.5-vl | 2-channel coarse | 2-channel fine | semi-duplex w/ ASR | full-duplex w/o ASR |

## C    Noise Augmentation for SFT

Noise sources include the DNS Challenge dataset (Dubey et al., 2023), RNNoise[7] , and random speech clips from Stage-1 post-training data. For each training sample, we add concatenated random noise segments to the listening channel waveforms. With probability 0.6, wave gain is applied to user utterances, scaling amplitudes within a range of -24 to +20 dB. We enforce a minimum final loudness of -40 dB. We compute $dB = 20.0 \times \log_{10}(\text{wave\_root\_mean\_square} + 1e{-}6)$. Noise clips are randomly scaled to (-70, -40) dB. Additionally, with probability 0.3, noise segments are replaced with silence.

Following Moshi, in the final stage, we also apply speech leakage augmentation by mixing the speaking channel back into the listening channel with probability 0.3, applying a random gain (0-0.2) and a random delay (0.1-0.5 seconds) to enhance robustness in microphone-based interaction.

## D    Rebuttal Revision

This section aims to address common concerns raised in the rebuttal stage.

### D.1    Controlled Comparison: Contiguous vs. Word-level

Because FLM-Audio relies on contiguous monologues beginning from the first post-training stage, conducting a fully controlled comparison against a word-level alignment strategy trained on the entire dataset would be prohibitively expensive in terms of computational resources. To provide a quick but informative sanity check, we perform an ablation-style comparison by processing 5% of our post-training data using a Moshi-like (Défossez et al., 2024) word-level alignment method (ASR-style, where each word-level token slightly lags behind its pronunciation). We train this Moshi-like model under the same configuration and run a complete pass over this 5% subset. At the end of training, we evaluate both models using ASR word error rate (WER) on LibriSpeech-clean and acc_norm on HellaSwag (Zellers et al., 2019). Results are shown in Table 7. We find that our contiguous strategy converges substantially faster than the word-level alternative, consistent with our observation that

---

[4]https://openslr.org/38/
[5]https://www.openslr.org/68/
[6]https://www.openslr.org/47/
[7]https://github.com/xiph/rnnoise

FLM-Audio achieves comparable or better performance with significantly less training data than Moshi.

**Table 7:** Ablation analysis: contiguous monologues vs. word-level alignment.

| Strategy | Fleurs-zh (WER ↓) | HellaSwag (Acc. ↑) |
|---|---|---|
| Contiguous (FLM-Audio) | 18.2 | 61.6 |
| Word-Level (Moshi-like) | 22.3 | 58.3 |

### D.2 OBJECTIVE STATISTICS FOR FULL-DUPLEX EVALUATION

We additionally compute objective metrics on a test set containing background noise and random user interruptions. The reported metrics include turn-taking accuracy and the per-minute deviation from ground-truth in pause, gap, and overlap durations, following the definitions in (Wang et al., 2025; Nguyen et al., 2023). As in most prior work, the training and test data are drawn from the same distribution, and these objective statistics quantify how effectively the model learns full-duplex dialog behaviors. Results are presented in Table 8.

**Table 8:** Objective evaluation on noisy test set with interruptions.

| Turn-taking Acc. ↑ | $|\Delta\text{Pause}|$ ↓ | $|\Delta\text{Gap}|$ ↓ | $|\Delta\text{Overlap}|$ ↓ |
|---|---|---|---|
| 0.98 | 1.9 | 0.9 | 2.3 |

