# OpenReview forum: "FLM-Audio: Natural Monologues Improves Native Full-Duplex Chatbots via Dual Training"
_ICLR.cc/2026/Conference — Submitted to ICLR 2026_

### Official Review · Reviewer_cLE2 · 2025-10-29

**Soundness:** 2
**Presentation:** 2
**Contribution:** 2
**Rating:** 4
**Confidence:** 4

**Summary:**

This paper introduces FLM-Audio, a 7B full-duplex dialogue model that can listen and speak simultaneously. By using “natural monologues” (continuous sentences with pauses) and a “dual training” strategy to align text and audio, it achieves more natural, low-latency conversations with higher response quality using less training data.

**Strengths:**

The presentation of the main idea is generally clear and well motivated.

**Weaknesses:**

(1) Limited core contribution. The main contribution of this paper is rather narrow, focusing primarily on modifying the padding strategy in Moshi. Instead of word-level alignment (i.e., padding to correct bit-rate mismatches between audio and text tokens), the paper proposes preserving sentence integrity by padding at the sentence level. While this is an intuitive enhancement to the Moshi architecture, the contribution remains confined to this single framework. Moreover, with the emergence of other full-duplex models such as dGSLM [1], NTPP [2], and LLaMA-Omni [3], it appears that the proposed approach may not generalize to these architectures, which limits the broader impact of the work.

(2) Lack of systematic empirical evaluation. Several key claims in the paper lack sufficient empirical evidence. For instance, the repeated claim that “FLM-Audio achieves equivalent response latency” is not substantiated by any quantitative benchmark or experiment. Furthermore, the paper does not describe how fairness in comparison is ensured. For example, does Moshi also undergo post-training and supervised fine-tuning under the same setup? Without such clarification, it is unclear whether the reported gains stem from architectural changes or differences in training. Additionally, while Table 1 lists many full-duplex models for comparison, most are not actually included in the experiments, leading to inconsistency between the discussion and the evaluation.

(3) Figure and table presentation. The figures and tables need refinement for clarity. Figure 1 aims to illustrate alignment strategy differences but does not clearly convey the intended comparison. It should be visually polished for better interpretability. Similarly, Table 3 lacks explicit labeling—readers cannot tell which metrics are shown, what the numerical values represent, or whether lower or higher scores are better. Such essential details should not be left only in the main text.

In conclusion, this paper, in its current form, is not yet ready for publication. The contributions need to be articulated more clearly and supported by systematic, well-documented experiments. Strengthening the empirical analysis and improving the clarity of presentation would greatly enhance the paper’s completeness and credibility.

[1] Generative Spoken Dialogue Language Modeling [TACL 2023]
[2] NTPP: Generative Speech Language Modeling for Dual-Channel Spoken Dialogue via Next-Token-Pair Prediction [ICML 2025]
[3] LLaMA-Omni: Seamless Speech Interaction with Large Language Models [ICLR 2025]

**Questions:**

(1) What is the latency of the proposed approach compared to the Moshi baseline?

(2) Does the Moshi baseline follow the same experimental setup as the proposed approach — including identical training data, fine-tuning, and post-training procedures? This reviewer could not find any related descriptions in the paper.

---

> ### Author Response · Authors · 2025-11-28
> **Thanks for your review**
>
> Thanks for your constructive comments and insightful opinions. We would like to respond to your concerns as follows:
>
> Weaknesses
>
> 1、We acknowledge that our empirical validation is conducted only within the Moshi framework. However, the core idea of natural (contiguous) monologues is broadly applicable to any framework that **intends to introduce an additional simultaneous textual channel to audio streams**, as it minimizes overhead from the text modality. Moshi is simply an advanced example of such architecture.
>
> We see no conceptual barrier preventing natural monologues—and the associated training paradigm—from being incorporated into frameworks like **dGSLM** or **NTPP**, which are currently textless and could potentially benefit from integrating a textual channel. Our success with Moshi suggests this direction is promising and may inspire extensions to both native full-duplex systems and textless frameworks, indicating potential broader impact.
>
>  The only incompatible cases are frameworks that already rely on contiguous text channels, such as TDM or semi-duplex approaches.
>
> 2、
> - The statement “FLM-Audio achieves equivalent response latency” holds because **both systems use exactly the same backbone model structure**, and the textual-channel design introduces **no additional overhead**. They generate the same audio length in the same number of decoding steps. This property is inherent to the architecture and does not require experimental validation.
> - We have already compared the post-training scheduling in Appendix B of the original paper, and the data we present in Section 3 clearly differs from that reported by Moshi. Actually, one of our key findings stated in the introduction is that different alignment strategies require different training paradigms. We agree that a controlled experiment directly validating the role of natural monologues is valuable, and this has been a common concern; please refer to our detailed update in the **response to all reviewers** and **Appendix D**.
> - In the revised version, we have expanded Table 3 to include more full-duplex (TDM) models.
>
> 3、We sincerely appreciate your suggestions and have improved the tables accordingly in the revised version. However, Figure 1 is not intended to illustrate alignment-strategy differences, but rather to compare the input sequence lengths between TDM structures and native full-duplex architectures.
>
>
> We hope these clarifications address your concerns. We appreciate your feedback and welcome further discussion. Thank you again for your valuable comments.

---

### Official Review · Reviewer_2jGm · 2025-10-31

**Soundness:** 2
**Presentation:** 2
**Contribution:** 2
**Rating:** 2
**Confidence:** 4

**Summary:**

The paper presents a full-duplex chatbot built on Moshi's codec and a Moshi-like parallel multi-channel framework. It introduces "natural monologue," a sentence-level alignment method to replace word-level alignment in Moshi, reducing degradation in language modeling and lowering data preprocessing costs. Additionally, it proposes a "dual" training paradigm that mixes TTS-style and ASR-style training data. Experiments demonstrate comparable human evaluation results with Qwen-2.5-Omni for full-duplex chatting tasks.

**Strengths:**

1. Full-duplex speech interaction is an important research topic.
2. The paper replaces Moshi’s word-level alignment with natural monologue (sentence-level alignment), significantly reducing data preprocessing costs.
3. Subjective evaluation is conducted for the full-duplex interaction task.

**Weaknesses:**

1. The novelty compared to Moshi is incremental, as it uses the same audio codec, with key changes being the shift from word-level padding to sentence-level padding and the mixed training data (TTS-style and ASR-style).
2. Table 3 lacks comparisons with common baselines such as Qwen2.5-Omni and Kimi-Audio.
3. Experimental design is flawed. For instance, DeepSeek-V3 refines responses for instruct-following data and is also used for LLM-score evaluation in full-duplex chatting, creating an unfair comparison with other baselines.
4. The paper does not report objective full-duplex metrics, such as turn-taking statistics in generated dialogues, including the number of turn-taking events and accumulated durations per minute compared to ground truth.
5. The natural monologue design is highlighted as the main innovation, but the paper lacks controlled experiments isolating its impact compared to Moshi-like word-level alignment.

**Questions:**

Please address the issues described in the Weaknesses section. Resolving these concerns could improve the paper’s evaluation.

---

> ### Author Response · Authors · 2025-11-28
> **Thanks for your review**
>
> Thanks for your constructive comments and insightful observations. We would like to respond to your concerns as follows:
>
> 1、We agree that our contribution is incremental; the novelty primarily lies in our data organization strategy and the training algorithms designed for full-duplex models.
>
> 2、We have updated Table 3 with additional results sourced from related papers.
>
> 3、We agree that the current "Instruct LLM-score" can potentially be problematic. For a more fair comparison, we conducted the following additional experiments:
>
> - **GPT-5 re-evaluation**:
>
>  Using GPT-5 as the judge, Qwen-2.5-Omni and FLM-Audio-SFT-full obtain **7.13** and **7.05**, respectively—again showing comparable performance.
> - **Human head-to-head evaluation**:
>
>  A graduate student evaluated 40 randomly sampled instructions from the test set in a **blind, pairwise comparison** (on the same task as in Table 5).
>  The human vote distribution for FLM-Audio (win / tie / lose) is: 12 / 14 / 14, indicating no significant difference between the systems.
>
> - **GPT-5 head-to-head evaluation**:
>
>  Using GPT-5 in the same blind pairwise setup, the vote distribution is: 10 / 17 / 13, which again confirms comparable performance.
>
> These results provide additional evidence supporting our claim. We are happy to update Table 5 accordingly if you agree these new experiments are more convincing.
>
> 4、This is a great suggestion. We have added several objective metrics in the new **Appendix D.2**.
>
> 5、This is a common concern. Please refer to our **response to all reviewers** and **Appendix D.1**.
>
> We hope these clarifications address your concerns. We appreciate your feedback and welcome further discussion. Thank you again for your valuable comments.

---

### Official Review · Reviewer_k7bv · 2025-11-03

**Soundness:** 3
**Presentation:** 3
**Contribution:** 2
**Rating:** 4
**Confidence:** 3

**Summary:**

The paper presents FLM-Audio, a 7B bilingual (English–Chinese) full-duplex spoken dialogue model built on a “native full-duplex” architecture that merges listening and speaking channels at each timestep. The authors identify a key limitation in prior work such as Moshi—word-level alignment between text and audio tokens—and propose “natural monologues,” continuous sentence-level text streams interleaved with <wait> tokens to mimic human cognition during speech. They further introduce a dual training paradigm, alternating between ASR-style and TTS-style supervision to better align asynchronous text and audio modalities. Experiments on ASR, TTS, and full-duplex dialog benchmarks show improvements over Moshi and comparable quality to Qwen2.5-Omni with significantly less training data.

**Strengths:**

1. Innovative reformulation of alignment in full-duplex systems via natural monologues.
2. Dual-format training effectively integrates both speech recognition and synthesis modes.
3. Extensive experiments with solid ablations demonstrate the importance of each training stage.
4. Competitive performance despite limited data (1M hours vs. 8M+ for Moshi).

**Weaknesses:**

1. Lack of deep analysis: no evidence is provided to explain why the method works beyond intuitive analogies to human cognition.
2. Weak empirical attribution: ablation results are limited and fail to isolate the effect of natural monologues versus other hyperparameters.

**Questions:**

1. Could you provide quantitative evidence that “natural monologues” improve language modeling capacity (e.g., perplexity, coherence scores)?
2. What are the exact latency, throughput, and memory metrics compared to TDM-based systems in real deployment?

---

> ### Author Response · Authors · 2025-11-28
> **Thanks for your review**
>
> Thanks for your constructive comments and insightful opinions. We would like to respond to your concerns as follows:
>
> 1、Your main concern is the lack of controlled experiments directly demonstrating that contiguous (natural) monologues outperform word-level alignment. This was also raised by several other reviewers. We have now provided new controlled results and additional discussion; please refer to our updated analysis in the **response to all reviewers** and **Appendix D**.
>
> 2、Our throughput of inference is 24.76 fps, which significantly exceeds the minimum requirement for the real-time streaming audio decoder (12.5fps). As described in the paper, the structural lower bound for latency in native full-duplex system is 80ms, but in order to avoid oversensitivity to short back-channels from user, we train the model to enter a silence state 0.5 seconds after detecting an interruption (Section 3.2.2). For normal queries, we train the system to begin responding 0.7–1.0 seconds after the user finishes speaking. Under this configuration, the average user-perceived latency is ~0.7 seconds.
>
> Memory metric: without any engineering optimization, on a single H100 (80GB), our system handles ~8000 contextual frames (≈ 640 seconds of audio). For comparison, a TDM-style version of our model (implemented by flattening listen/speak/text channels as in Figure 1) supports only ~2200 frames. We note that latency can be adjusted through the latency configuration used during training-data construction, and both throughput and memory efficiency can be further improved with engineering optimizations.
>
> We hope these clarifications address your concerns. We appreciate your feedback and welcome further discussion. Thank you again for your valuable comments.

---

### Official Review · Reviewer_apT6 · 2025-11-07

**Soundness:** 4
**Presentation:** 3
**Contribution:** 3
**Rating:** 8
**Confidence:** 4

**Summary:**

This paper describes the design and implementation of a transformer-based AI model named FLM-Audio for full-duplex spoken conversation in English and Chinese. Full-duplex means that the user and the system can interrupt each other and speak over each other. The system begins with a textual LLM and does additional training to enable full-duplex speech.

The input is a temporal stream of tokens. In time-division multiplexing (TDM), output audio, input audio, and output text (called monologue) are interleaved sequentially. In contrast, in native full duplex, each token is the concatenation of one output audio token, one input audio token, and one output text token (see Figure 1). The first main innovation of FLM-Audio is visible in line 173 (Figure 2): the text tokens of one utterance (a sentence or similar) are contiguous. Audio requires many more tokens than text, so after each text utterance. "wait" tokens are concatenated with the audio tokens. In contrast, in Moshi, text tokens are individually aligned with audio, so "pad" tokens are inserted between text tokens of the same utterance.

The second main innovation of FLM-Audio, described in Section 3, is in the specifics of the post-training that converts the original text-only LLM into a spoken dialog model.

**Strengths:**

The innovations of this paper are sensible. They have been fully implemented, and the experimental results are good. The innovations are building blocks that other teams will be able to mix and match in the future when creating even better spoken dialog systems.

**Weaknesses:**

The innovations are not highly surprising, and they are not the result of any major new insight or empirical discovery.

There is hardly any experimental comparison to the closed-source state of the art.

The terminology "natural monologues" makes an implicit claim that is too strong. "Contiguous text utterances" would be more precise. Also, "human-like" instead of "humanoid" would avoid the robot connotations of "humanoid," which are not appropriate here.

A note on LLM usage: It looks very likely that the authors used AI to make the text conform to the conventions of standard English. Line 498 says "We hope these efforts facilitate the community in implementing our methods and achieve results consistent with our conclusions." In this sentence, the non-standard direct object of "facilitate" and the unintended literal meaning of "We hope these efforts achieve results consistent with our conclusions" reveal what the writing might have been like before improvement by AI.

The singular-plural mismatch in "Natural Monologues Improves" in the title is other evidence of the lower quality of the English before AI improvement.

**Questions:**

Quantify the resources used for training and inference, and the latency achieved.

Line 45: What infrastructure is assumed for the numbers 2 and 45 seconds? What is the cost of this in dollars or other units? What is the % improvement achieved by FLM-Audio?

Please say more about the state of the art in the entire field of spoken dialog systems:
- Are there any alternatives to transformers, and to post-training a text-only model?
- Are current systems better or worse than humans? In what ways, and why?
- Why should we believe that the innovations here are still valuable when building on top of a text-only model that is much larger and better?

---

> ### Author Response · Authors · 2025-11-28
> **Thanks for your review**
>
> Thanks for your constructive comments and positive opinions. We would like to respond to your concerns raised in weakness and questions as follows:
>
> Weakness:
>
> 1、Comparison to closed-source state-of-the-art systems:
>
> Unfortunately, we cannot access closed-source systems such as GPT-5 in a true live full-duplex setting, which makes meaningful human evaluation difficult—especially for response latency, where web latency and API buffering strongly distort the results.
>
> For a **non-strict upper-bound comparison**, we tested GPT-5’s voice mode on a 50-sample subset of our evaluation set purely to provide an upper limit for instruct-following and partial human evaluation:
>
> |Model|LLM Instruct Score|Helpfulness|Robustness|
> | :-------- | :------- | :-------- | :------- |
> |GPT-5 Voice Mode|7.6|7.8|8.1|
>
> These results mainly reflect GPT-5’s language modeling abilities and its robustness to input noise, rather than full-duplex performance.
>
> 2、Writing improvements:
>
> We sincerely appreciate your suggestions regarding clarity and writing quality. We have incorporated the recommended revisions in the rebuttal-updated version of the manuscript.
>
> Questions
>
> 1、The numbers we cite are directly taken from the referenced works: 2 seconds is reported by MiniCPM-duplex, and 45 seconds is reported MiniCPM-o-2.6, both being representitive TDM systems. Their higher latency arises because they re-encode audio **every 2 seconds**, whereas our token stream operates at 12.5 steps per second **(80 ms)**. The native duplex design of Moshi and FLM-Audio supports audio context of up to 10 minutes in a single 80G GPU. These improvements primarily enhance **user experience** rather than reduce overall compute cost. TDM systems can prune context or use engineering optimizations to lower cost, but still cannot match the responsiveness of native duplex models.
>
> 2、More on the related work:
>
> - While alternatives to the Transformer architecture are rare, there do exist alternatives to text-only pretraining. For example, MiMo-Audio (https://github.com/XiaomiMiMo/MiMo-Audio) is trained from scratch jointly on text-only, speech-only, and speech-text corpora, using more than 100× our data volume (>100M hours).
> - Current models still lag behind humans in handling multi-speaker overlaps and personalization. This reflects the inherent difficulty of determining the active speaker and their intent in mixed audio— the well-known cocktail party problem.
> - Regarding whether larger LLMs could make word-level alignment competitive: it is possible that sufficiently large models may maintain linguistic consistency even with word-level alignment. However, contiguous monologues retain key advantages:
>   - **Avoiding error cascading** from imperfect timestamp alignment or smaller ASR models used during data preprocessing.
>   - **Faster convergence** in post-training for the same perplexity, reducing computational cost—especially important when scaling model size.
>
> We hope these clarifications address your concerns. We appreciate your feedback and welcome further discussion. Thank you again for your valuable comments.

---

### Author Response · Authors · 2025-11-28
**Response to All Reviewers**

We sincerely thank the reviewers for their thoughtful and constructive feedback. Your comments have greatly improved the clarity and overall quality of our paper. Below we summarize the major revisions and provide clarifications addressing common concerns.

**Paper Modifications**
- We updated several terminologies (e.g., natural → contiguous) in accordance with reviewer suggestions.
- The presentation of evaluation metrics in the tables has been refined for improved readability.
- We added results from state-of-the-art semi-duplex / TDM systems in Table 3.
- We added **Appendix D**, which includes:
  - **D.1**: A controlled comparison between contiguous monologues and word-level alignments.
  - **D.2**: Additional objective statistics.
- A one-minute dialog demonstration (with English transcripts) is provided in the **supplementary materials**.

**Response to Common Concerns**

A major common concern is a controlled study showing the performance improvement can be attributed to the proposed contiguous monologue under controlled setting. Because FLM-Audio relies on contiguous monologues beginning from the first post-training stage, conducting a fully controlled comparison against a word-level alignment strategy trained on the entire dataset would be prohibitively expensive in terms of computational resources. To provide a quick but informative sanity check, we perform an ablation-style comparison by processing 5% of our post-training data using a Moshi-like word-level alignment method. For details, please refer to Appendix D.

---

### Meta-Review · Area_Chair_JYyk · 2026-01-08

**Summary:**

1. Incremental Contribution and Limited Novelty:
The reviewers noted that the paper’s main contribution—shifting from word-level to sentence-level alignment for full-duplex systems—is incremental, mainly improving upon the Moshi framework.

2. Lack of Sufficient Empirical Evidence:
A significant concern was the lack of comprehensive empirical analysis to substantiate key claims.

3. Weak Experimental Comparisons:
The reviewer pointed out that comparisons to other state-of-the-art models were either absent or incomplete.

4. Clarity and Presentation:
The reviewer highlighted issues with the clarity of some figures and tables, specifically pointing out that Figure 1 was not sufficiently clear in comparing alignment strategies, and Table 3 lacked labels and needed refinement.

**Reviewer Concerns:**

Concerns Still Outstanding:

1. Lack of Controlled Experiments:
While the authors provided more details on the comparisons, the rebuttal still lacks a fully controlled experiment that isolates the impact of "natural monologues" from other factors such as training and data preprocessing.

2. Quantitative Evaluation of Natural Monologues:
The rebuttal addresses the lack of direct quantitative evidence for the impact of "natural monologues" by including new experiments, but it still doesn’t fully address the specific request for quantitative metrics like perplexity or coherence scores that could demonstrate their improvement in language modeling capacity.

3. Comparison to Other Full-Duplex Systems:
The comparisons with other full-duplex systems like dGSLM, NTPP, and LLaMA-Omni were insufficient. The authors’ response that their method could generalize to these systems is helpful, but it would be stronger if they could provide direct experimental comparisons with these models, particularly given the growing body of work in the area.

**Reviewer Scores:**

None

---

### Decision · Program_Chairs · 2026-01-26

Reject